# A Review on Microscopic Visual Servoing for Micromanipulation Systems: Applications in Micromanufacturing, Biological Injection, and Nanosensor Assembly

**DOI:** 10.3390/mi10120843

**Published:** 2019-12-02

**Authors:** Xiaopeng Sha, Hui Sun, Yuliang Zhao, Wenchao Li, Wen J. Li

**Affiliations:** 1School of Control Engineering, Northeastern University at Qinhuangdao, Qinhuangdao 066004, China; shaxiaopeng@neuq.edu.cn (X.S.); sunhui@stumail.neu.edu.cn (H.S.); liwenchao@neuq.edu.cn (W.L.); 2Department of Mechanical Engineering, City University of Hong Kong, Kowloon, Hong Kong 999077, China

**Keywords:** micromanipulation, calibration, autofocus, visual servoing control, microlens

## Abstract

Micromanipulation is an interdisciplinary technology that integrates advanced knowledge of microscale/nanoscale science, mechanical engineering, electronic engineering, and control engineering. Over the past two decades, it has been widely applied in the fields of MEMS (microelectromechanical systems), bioengineering, and microdevice integration and manufacturing. Microvision servoing is the basic tool for enabling the automatic and precise micromanipulation of microscale/nanoscale entities. However, there are still many problems surrounding microvision servoing in theory and the application of this technology’s micromanipulation processes. This paper summarizes the research, development status, and practical applications of critical components of microvision servoing for micromanipulation, including geometric calibration, autofocus techniques, depth information, and visual servoing control. Suggestions for guiding future innovation and development in this field are also provided in this review.

## 1. Introduction

Micromanipulation or microassembly techniques refer to the processing or manipulation of tiny objects, which are generally between millimeters and nanometers in size. They are mainly used in biological engineering, microdevice processing or assembly, medical engineering and other fields of fine operation [1]. MEMS (microelectromechanical systems) [2,3], MOEMS (micro-optoelectromechanical systems) [4], BioMEMS (biological microelectromechanical systems) [5,6,7,8], and other similar microsystems are generated as a result of the multidisciplinary interactions between sensors, precision machining, biological engineering, microelectronics and precision measurement technologies. The objects of these systems have small volumes, weak structures, light weights and low rigidities, and the forces applied to them should not be too large. It is necessary to precisely position the object and finely adjust the shape and posture of the robot to avoid the operating object being over-contacted by forces or inaccurate operating positions. Therefore, it is quite important to add visual feedback information to a micromanipulation system [9,10]. The integration of visual information will also build a technical foundation for the automation and intelligence of the micromanipulation system. It is of great significance for safe, stable and accurate operations of tiny objects.

## 2. Application

In recent years, many micromanipulation systems have been developed for different applications, such as genetic engineering by injecting DNAs into cells [11,12], assembling nanomaterials to form nanosensors for healthcare/environmental safety applications [13,14], assembling vascular-like microtubes [15,16], and microcomponents for mobile phone applications, etc. These applications have greatly promoted the development of automation science and engineering. Some examples of the key applications are shown as follows.

### 2.1. MEMS

A forecast indicates that the market size of MEMS and sensors will increase with a composite growth rate of 13% [17]. The rapid increase in the demand for MEMS has driven governments and research institutions to involve numerous researchers and other resources in developing multifunctional MEMS devices for their respective applications [18,19]. MEMS are considered to be made of multiple independent components, which can be manufactured by using different precision machining methods and different materials, and then these components can be put together to create the required systems via microassembly or micromanipulation and used in different fields [20,21]. The performance of this process increasingly relies on microvisual servoing. Figure 1a shows a good example of microassembly using SEM (scanning electron microscope). With the fast development of MEMS, the focus of this field has shifted from microdevices or single-function parts to fast and precise integration of complex systems. Meanwhile, the increasingly miniaturized objects, the complex shapes and the more diverse processing and manufacturing trends have made it difficult to supervise and control micromanipulation effectively. Manual micromanipulation and microassembly are clearly unable to satisfy the upcoming requirements for mass industrial production [22,23]. Therefore, microvisual servoing techniques will play a more important key role in the development of this field.

### 2.2. Nanoscale Assembling

Nanomaterials, due to their small size and outstanding properties, are considered to be promising units for nanoelectronic devices [24]. Carbon nanotubes (CNTs) have been widely studied due to their electrical, mechanical, and chemical properties. Figure 1b shows a 3D CNT assembly. Due to CNTs’ miniature size and tendency to cling together in nature, connecting, aligning, and isolating processes of CNTs are difficult. Atomic force microscopy (AFM) is typically used to manipulate each nanosized tube one-by-one. However, it is time-consuming and unrealistic when considering batch production processes. In order to perform a precise and efficient batch manipulation process for CNTs, automated CNT microspotting systems are developed based on the technique of electric-field assist assembly (i.e., dielectrophoresis). An automated CNT microspotting system generally includes a PC with a video/image acquisition card, a CCD (Charge Coupled Device) camera, a microscope, a micromanipulator, and a dielectrophoresis microdevice. In such a system, the microscopic vision will guide the system to complete the assembly task, and this feasible batch manufacturable method will dramatically reduce the production costs and production time of nanosensing devices and potentially enable fully automated assembly of CNT-based devices [13]. Figure 1c shows connections of bundled CNTs for sample pairs of microelectrodes on a substrate after one spotting cycle of CNT dilution.

### 2.3. Genetic Engineering

Nowadays, biological micromanipulation is an important approach in biomedical engineering, which concerns the operation of biological entities such as positioning, gripping, injecting, cutting, and fusion, etc. In particular, the single cell is the smallest unit of biological things (typically, around 10–500 μm size) and acts as the basic component of life [12,25]. Therefore, biological cell micromanipulation has gained extensive interest from both academia and industry in the past two decades. Traditionally, biological cells are manipulated manually by an operator using the visual information provided by an optical microscope. However, manual operation suffers from low efficiency, low success rate, and low repeatability. Moreover, the long-time operation will cause fatigue to the human operator [26,27,28].

Alternatively, a robotic micromanipulation system delivers actuation, sensing, and control capabilities to allow precision positioning, gripping, and assembly of micro-objects. In an automated microinjection system, vision is the primary source of feedback for control and visual servoing. Visual feedback in automated microinjection is generally used in two ways: (1) detection and tracking of the object with derivative functions including cell modeling, injection process tracking and 3D cell reconstruction for user interface purposes; (2) height estimation, i.e., the kinematic relationship between the cell and injection pipette [29]. Figure 2 shows the main parts of an automated microinjection system.

## 3. Microscopic Visual Servoing Technologies for Micromanipulation Systems

With robotic devices playing a central role in the development of micromanipulation systems, the physical size of the objects to manipulate is becoming smaller and smaller, which requires higher standards of accuracy, automation and visualization to be put in place in micromanipulation systems.

Considering the characteristics of the objects, the force should be applied correctly and limited to a reasonable level so that the objects are not damaged in the process. This requires precise determination of the object’s position and the microrobot’s shape and posture. As such, microvision can be used as a non-contact, high-precision method to monitor in real time the dynamics, such as posture and movement, of microdevices. Therefore, microvision has proved to play a critical role in micromanipulation systems. Its ability to incorporate various pieces of microvision information in one integrated package also lays a solid foundation for the automation and intelligence of micromanipulation systems, which is instrumental to the achievement of safe, stable and accurate micro-object manipulation. This integrated system is called a microvision system [24,34,35,36], as shown in Figure 3a. However, at present, there are still many problems to be solved in microvision systems. Furthering the research on microvision systems has become an urgent task for the development of micromanipulation systems.

The experimental platform involves *system calibration, autofocusing*, *depth information extraction*, and *microscopic visual servoing control*. These form the general process of the microscopic visual serving system, which is illustrated in Figure 3b.

### 3.1. System Calibration

#### 3.1.1. Definition of Calibration of Microscopic Vision

The calibration of a micromanipulation system provides appropriate target space information for subsequent servo operations, and its precision has a direct impact on the precision of the entire micromanipulation system. Developing calibration techniques suitable for microscopic vision is an important step toward the automation of micromanipulation systems, from laboratory to industry [37].

The camera imaging model is a simplification of the geometrical relationship of optical imaging. There are a variety of imaging models depending on the type of camera used and the specific use case. Generally, they can be divided into the pinhole imaging model, nonlinear lens model, and linear approximate model. The pinhole imaging model, a type of linear model, is an ideal basic option for developing camera calibration techniques. However, its applicability is compromised by its limited amount of light and low imaging speed. As for the nonlinear lens model, factors like the design complexity and processing capacity of a lens can cause varying errors, which inevitably results in distortion in the obtained images. Correcting the image based on the distortion model is a necessary step in subsequent activities. This model is called a nonlinear model considering the distortion in lens imaging. Although the perspective projection has taken into account the pinhole model and lens distortion compensation, it is still a nonlinear mapping, and massive computations will be involved when it comes to actual calculation. This model may not produce any solution if the perspective effect is not obvious. Hakan et al. from Turkey implemented a new online calibration method in a micromanipulation platform. This method used CAD template matching to get a correspondence, assuming parameters of three consecutive frames are invariant to obtain solutions of the three corresponding points which satisfy the calibration equation, and finally evaluated various parameters of the system via parameter estimation [38].

Currently, the ordinary calibration methods are Faugeras’ method [39] and Tsai’s method [40]. These two methods are classical methods of stereoscopic targeting. Faugeras’ method calibrates the internal and external parameters of the current camera model, and Tsai’s method takes lens distortion into consideration. The most widely used is Zhang’s plane template calibration method [41].

#### 3.1.2. The Process of Calibrating a Microstereoscopic Visual System

① Calibration of a binocular camera

The main job in calibrating a microstereoscopic visual system is to determine the structure model of a binocular camera. The idea is to calculate the structure parameters of the left camera relative to the right one based on the two optimized microscopic visual models, after the calibration of the two cameras. The position coordinates of objects are determined by the structure model and camera parameters, then the 3D coordinates of target objects relative to the camera coordinate in 3D space are obtained. These coordinates can be used for subsequent coordinate transformation and micropositioning control.

② Calibration process

The process of calibrating a binocular camera is shown as Figure 4. Firstly, we initialize the binocular target data, locate and load the binocular images, and then load the calibration plate description file and initial parameters of the cameras. Meanwhile, the image pairs of the calibration plate in the left and right cameras are collected, and then the location information of the calibration plate area and all calibration points is extracted. Then, the data are loaded into the binocular array and the binocular image is collected circularly. The binocular camera calibration will be carried out after reading the number of set images. Finally, the internal and structural parameters of the binocular camera are output to correct the polar line.

### 3.2. Autofocusing

#### 3.2.1. Development Process of Autofocusing

Because of the short field depth of a microscope, the images acquired when the micromanipulator and controlled objects move along the optical axis of the microscope may become blurred. Relying on the operator’s subjective judgments alone may produce significant human errors, which can be a fatal flaw in situations that require absolute measurement accuracy. Finding a fast and accurate way to obtain the clearest imaging position of a target has become a major task in the study of micromanipulation systems [42].

Image definition and positive focus location search are the two basic technical problems in autofocusing technology based on image processing. For an image sequence of the same object, the technique to determine which location is the sharpest (the target is on the focal plane) is called definition evaluation function. The main function of definition evaluation function is used to describe the focusing degree of the current image in numerical form. Ideal autofocusing is usually achieved through definition evaluation function (focusing function), focus location search and a determined extremum search strategy. Figure 5 shows how image processing-based autofocusing technology works.

Currently, there are three types of definition evaluation function. The first type is based on the spatial domain, with variations such as the SMD (Sum of Modulus of gray Difference) function, the EOG (Energy of gradient) function, the Krish function, the Tenengrad function, the Brenner function and the Laplace function. The second type is based on the frequency domain, and its variations include the Fourier transform and the discrete cosine transform. The third type is based on image statistics and relativity, with typical examples including the Menmay function, the Range function and the Variance function [43].

In 1976, Brenner et al. at Boston Medical Center, USA first proposed an evaluation function based on grayscale gradient. Later, it has been called the Brenner function [44]. In 1985, Groen et al. proposed a variance function that had better performance than the grayscale gradient evaluation function in terms of sensitivity, stability, and calculation speed [45]. In 1987, the Tenengrad function was put forward by Krotov et al. This function is based on the Sobel operator to process an image, and then the horizontal and vertical gradients in the image are calculated. The effect is very obvious after focusing [46]. In 1993, Firestone et al. proposed a non-gradient evaluation function, the Range function [47]. This function employs all effective information in the image, making the study of definition evaluation functions independent from the traditional automatic technology, and since then, definition evaluation functions have been diversified.

Several achievements have also been made in research on algorithms for the location of positive focus. At present, autofocusing methods such as hill climbing, binary research and Fibonacci research are commonly used. “Focusing Techniques”, published by Subbarao et al. in 1993, had a great influence on the field of autofocusing. In this paper, the formation principle of focusing and defocusing is expounded in detail from the angle of optics, frequency and energy [48]. At present, the commonly used autofocusing methods include hill climbing, two-point search and Fibonacci search. For micromanipulation systems, many researchers have adopted a variety of focusing methods to put forward some image-based autofocusing technologies. Lee et al. from KIST (Korea Institute of Science and Technology), South Korea adopted a coarse/fine two-level focusing method for autofocusing of a microscopic visual system. In the coarse adjustment process, the focal plane of the focused area was initially determined by using global images or images at low optical magnification. In the fine tuning stage, the focal plane was accurately determined by a local image or high optical magnification image [49].

#### 3.2.2. Existing Problems of Imaged-based Autofocusing Technology

Designing and selecting a focusing evaluation function is key to achieving autofocusing in micromanipulation systems. How to select an appropriate one for practical application from the diverse array of focusing functions has become a key concern. Currently, objective quantitative criteria are absent for the selection of autofocusing functions. Quantitative evaluation indicators can not only be used in the selection of optimal functions, but also provide a theoretical basis for the design of new focusing functions.

Despite the extensive research on quantitative evaluation indicators, there still exist some problems: (1) The evaluation of focusing functions still remains at the qualitative level, without employing any objective quantitative indicator. The evaluation is mostly performed by observing the focusing curve to determine the performance of a function, but how well it performs cannot be reflected by numerical values; (2) Some of the evaluation indicators are biased, and important indicators such as the steep region width of a focusing function, which directly affects the range for focus step selection, have not received much attention; (3) In the past, the evaluation of focusing functions was mostly based on specific images, ignoring the effect of image content on the performance of focusing functions. The fact is, however, image content has a significant effect on the shape of focusing function curves. For the same focusing function, rich image content will lead to a steep focusing curve, and scarce image content will often lead to an insignificant steep value. Sometimes it is even impossible to find the focal plane. Therefore, image content is another factor to be considered in evaluating autofocusing functions.

### 3.3. Depth Information Extraction

Most microvisual systems are based on image-based visual servoing [50]. The mapping relation between the differential of an image feature point and the camera translation speed and rotation speed is called the image Jacobian matrix, which plays an important role in designing control algorithms. This matrix reflects a mapping transformation relation from the image feature point space to the robot operation space. How to obtain the image Jacobian matrix is a key issue in the image-based visual servo (IBVS). For monocular visual servo systems, the following image Jacobian matrix is generally used:(1)$$\dot{m}=J(m,Z)\omega$$
$\omega ={\left[\begin{array}{cccccc} T_{x} & T_{y} & T_{z} & \omega_{x} & \omega_{y} & \omega_{z} \end{array}\right]}^{T}$ is the velocity of the robot terminal actuator, $m={\left[\begin{array}{cc} x & y \end{array}\right]}^{T}$ is the image plane coordinates of a feature point, and J(m,Z) is the image Jacobian matrix, as shown in Equation (6).

In Equation (1), Z represents the feature points’ depth information. However, for monocular cameras, the depth information cannot be directly obtained by only one image.

Because microvision has a small field of vision and short depth of focus, how to obtain the depth information of the target in a micromanipulation system is a key problem that has plagued the development of micromanipulation for a long time [51]. At present, the microvisual part of a micromanipulation system mostly adopts the monocular model to collect visual feature points as visual feedback information. However, the monocular model has insuperable shortcomings: the deep blind area problem and the single point target control problem.

#### 3.3.1. The Blind Area of Depth Information in the Monocular Model

In monocular imaging, the imaging formula for the x-axis direction is:(2)$$x_{1}=f\frac{X_{1}}{Z}$$
where f is the focal distance, $X_{1}$ is the coordinates of the object point on the x-axis, $x_{1}$ is the coordinates of its image on the x-axis, and Z is the depth value of the point.

In order to study the imaging relationship between the depth information and the object, we consider Z as a variable, f and $X_{1}$ as constants, and take the derivative of (1):(3)$${\dot{x}}_{1}=fX_{1}\frac{-\dot{Z}}{Z^{2}}$$

Transfer (2) into incremental form:(4)$$\Delta x_{1}=fX_{1}\frac{-\Delta Z}{Z^{2}}$$

The imaging sensitivity $\eta_{x1}$ of the x-axis in the depth direction is:(5)$$\eta_{x1}=\left|-\frac{\Delta x_{1}}{\Delta Z}\right|=\frac{f\left|X_{1}\right|}{Z^{2}}$$

Similarly, this result is also suitable for the y-axis direction.

According to the above formulas, the change in the depth information Z causes the changing rate of the x(y)-axis imaging information to be proportional to $X_{1}$($Y_{1}$) (the distance from the x(y)-axis to the object point), that is, the farther the imaging point is away from the optic center, the higher the sensitivity, and vice versa. The sensitivity of the imaging point near the optic center is approximately zero ($\eta_{x1}\approx 0$).

From the above analysis, in a monocular visual imaging system, the volume of the depth information provided by imaging points is variable, and it has a function relationship with the location of imaging points. The farther the imaging point is away from the image center, the more depth information is provided by the image points, and vice versa.

#### 3.3.2. Single Point Target Control Problem of the Monocular Model

For a monocular visual system, if the target is a “dot” signal, it can be seen from the imaging formula that there will be many targets to get the imaging point, which are in a straight line. If the “point” moves along the straight line, the location of the “dot” imaging is almost immovable, which means no new target feedback information can be provided. So, a monocular lens cannot control single point targets.

#### 3.3.3. The Problem of the Depth Information Z in the Monocular Model

Currently, the following image Jacobian matrix is used in most visual servoing systems:(6)$$J(m,Z)=\left[\begin{array}{cccccc} -\frac{f}{Z} & 0 & \frac{x}{Z} & \frac{xy}{f} & \frac{-(x^{2}+f^{2})}{f} & y\\ 0 & -\frac{f}{Z} & \frac{y}{Z} & \frac{-(y^{2}+f^{2})}{f} & -\frac{xy}{f} & -x \end{array}\right]$$

In Equation (6), Z is the actual depth of target feature points. The significant drawback of the above model is the existence of the depth parameter Z, which is a variable. When the target depth is known, it can be successfully controlled. However, when the depth changes, it will cause trouble for the system control. The primary cause of this problem is that the single lens plane graph cannot directly measure the depth of the target object, which is an inherent shortcoming of a monocular visual system.

There are three ways to obtain the depth information of a microvisual system, which are shown in Table 1.

### 3.4. Microscopic Visual Servoing Control

In the field of robotics, the visual servo is defined as the visual information acquired by the visual sensor as feedback information to control the relative end pose of the robot to the reference coordinates or reference features [61]. The use of a camera in a robot control loop can be performed with two types of architecture: eye-in-hand and eye-to-hand. With the camera fixed on the microscope, microvisual servoing almost uses eye-to-hand to obtain the images that have been magnified by a microscope [62].

According to the visual feedback information used in the visual servo control process, the visual servo control can be divided into the position-based visual servo (PBVS) [63] and image-based visual servo (IBVS) [64,65,66]. The specific classification is shown in Table 2.

The position-based visual servo control is also known as the 3D visual servo. First of all, the visual task is defined in cartesian coordinates. The pose of the robot end-effector relative to the desired feature point of the target object is obtained according to the image information, and the relative pose is generated by comparing the current pose of the target with the desired pose. The motion command of the robot given by the relative pose is transmitted to the robot motion controller, and then the robot is controlled to move [67].

An automated microinjection system with high productivity for human cells with small size was proposed by D. Sun, who used the vision-based position-tracking method to recognize and position the target cells automatically [68]. Xiao. S. designed a visual servo feedback controller for a novel large working range microassembly manipulation system, and used the Fourier descriptors to identify and recognize the gripping fingers and object according to the silver wire rod for position information [69]. Brahim Tamadazte [70,71] from the FEMTO-ST Institute in France established a micromanipulation system as shown in Figure 6a (1). The system is placed on an anti-vibration platform to keep it in a controlled environment. The microrobot system is composed of two subsystems with five degrees of freedom. Figure 6a (2) shows how a microassembly task is completed. Meanwhile, Antoine Ferreira et al. proposed an automated micromanipulation workcell for visually servoed teleoperated microassembly assisted by virtual reality techniques. It is shown in Figure 6b. The micromanipulation system is composed of two micromanipulators equipped with microtools operating under a light microscope, and visual servoing techniques are applied for efficient and reliable position/force feedback during the tasks. For the imprecisely calibrated microworld, a virtual-microworld-based guiding system is presented. It is exactly reconstructed from the CAD-CAM databases of the real environment being considered [72].

The image-based visual servo is also called the 2D visual servo. The image feature is directly used as feedback information. By calculating the characteristic error of the current feature and the desired feature, which is present in the images as a directly control amount, it is loaded into the visual servo controller, and then the servo controller is fed back to the robot as a torque to control the robot motion.

Masatoshi Ishikawa et al. developed a microvisual manipulation system for operating a moving cell in 3D space [73,74], which is shown in Figure 7c. Based on the experimental equipment, the high-speed tracking and aggregation of the motor cells were completed through spot stimulation, and fast tracking of ascidians’ sperms was studied by using high-speed visual feedback with a frame rate of 1 kHz. A novel high-speed microrobotic platform that realizes long-time tracking and stimulation of a free motile microorganism in a microfluidic chip was proposed by Ahmad et al. To realize real-time target tracking, the block diagram of the visual servo controller is shown as Figure 7a, including the pixel pitch, focal length, and illumination control for the adaptive tracking. A simple image processing method was used, which utilizes the small spatial difference between two consecutive frames. The platform could successfully track targets that move with a velocity of up to 10 mm/s [75]. Brahim Tamadazte’s team studied a microvisual servoing algorithm based on the image gradient and optical flow technology. They also studied the location control method of multiview and monocular microvisual servoing based on this micromanipulation system [76]. Songlin Zhang et al. proposed a robust visual detection algorithm with the help of a microscopic visual servoing microinjection system, to determine the heart position of zebrafish from different zebrafish’s gestures. Further, the automatic rotation of cells was studied. Experiments showed that the successful rotation rate of the z-axis was about 94%, and that of the x-axis rotation was 100%. The system is shown in Figure 7b [77,78,79].

## 4. Forecast and Discussion

(1)Real-time image processing remains a major obstacle to visual servoing control. In order to improve the precision and speed of microscopic visual control, it is essential to develop more concise and efficient image recognition algorithms in addition to selecting high-quality image processing hardware.(2)At present, visual servoing systems are used to collect the 2D information of end effectors and manipulated objects. In order to meet a wider range of manipulation needs, some researchers have studied 3D data field visualization and real-time video capture and applied them in combination with virtual reality technology to build 3D object modes, allowing micromanipulation tools to achieve better Z-orientation location and quantitative control. However, because 3D reconstruction takes a long time to complete, it has been difficult to really apply this technology to micromanipulation control that requires very high real-time performance. How to eliminate this bottleneck is an issue facing researchers.(3)The contradiction between the workspace and resolution of a micromanipulator needs to be solved. In recent years, many scholars have utilized combined macro- and micromotion systems to achieve large-stroke, high-precision motion and location, but the control process is very complex. So, designers have been striving to find a way to design micromanipulators with a compact structure, a large workspace, high motion resolution, high bandwidth and high compliance.(4)Because of the constraints in the microworld and the various congenital defects in a single sensor, it is urgent to develop a new type of high-precision microdisplacement and microforce sensor. With the rapid development of micro- and nanotechnologies, some researchers have conducted preliminary studies on linear nanoservo motors in an attempt to integrate displacement sensors in them, and some results have been achieved; however, further research is needed.(5)In the microworld of manipulated objects, the kinematical and mechanical characteristics are different from some existing physical laws. In addition to gravity, buoyancy, flowage, Brownian motion, Van der Waals’ force and electrostatic force should also be taken into consideration. Researchers also need to work toward achieving free grapping and spontaneous detachment.(6)The micromanipulation control theory needs to be discussed further. A robotic micromanipulation system is a nonlinear system that is highly complex due to the difficulty in transferring accumulated errors, achieving real-time detection of micropositions and gestures, establishing accurate model design control strategies, and obtaining precise hand error signals for feedback control. Therefore, steady control precision of the system’s micromotion is hard to achieve (with poor robustness). Furthermore, how to build effective intelligent control algorithms has become a popular focus of interest for researchers.(7)The accuracy of machining and assembly is lower than the overall average of the system, which leads to difficulty in system calibration. The error of the conversion from each subsystem to the reference frame, and the random errors caused by temperature, vibration and creep add to the complexity in off-line calibration. In fact, it is impossible to achieve accurate and static calibration with a micromanipulation system alone. The calibration problem can only be solved by combining geometric calibration with an intelligent control and self-learning function.(8)Visual sensors are mainly used for the perception of noncontact global geometric information, which is easily affected by light and other factors. Contact sensors, such as force sensors, however, are a more suitable, reliable choice for the perception of contact force and other local information. Therefore, how to integrate the various sensor information to improve operational efficiency and accuracy has become an important focus of research.(9)At present, a time-efficient, non-invasive, environmentally compatible and high-throughput optical microscopy technique called scanning superlens microscopy (SSUM) has been proposed for large-area, super-resolution imaging and structural information acquisition. This microscopy operates in both non-invasive and contact modes, with 200 times the acquisition efficiency of an atomic force microscopy [90,91,92,93]. It enables large-area observation of live-cell morphology or sub-membrane structures, with sub-diffraction-limited resolution demonstrated by observing biological and non-biological objects. Figure 8 shows the non-invasive scan imaging of a mouse myoblast cell (C2C12) and a human breast cancer cell (MCR-7). Figure 9 shows the non-invasive in vivo light-sheet imaging of a mouse head using oblique NIR-II (Near infrared- II) LSM (Light-sheet microscopy). Therefore, using microlenses to enhance imaging and thus allow observation beyond the diffraction limit is another technique that may drastically change the visual servoing implementation for microscale/nanoscale manipulations in the near future.

## 5. Conclusions

Recently, a visual servoing technique has been actively introduced to the bioimaging field to take a single cell-level florescent image of a moving target [94], and the application of microscopic vision technology in the automatic detection of cancer cells has been researched [15,95,96]. The threat to human health from cancers is becoming increasingly serious, and drug susceptibility tests on clinical cancer patients mainly rely on manual labor, with a low degree of automation. Automating the process of primary cancer cell detection will potentially have tremendous economic benefits and social significance. Therefore, the application of microscopic vision technology to detect cancer cells for personalized medicine has a positive effect on human health and disease treatment.

With the advancement of microvision/nanovision, image processing, pattern recognition and robotic technologies, machine vision has been widely applied to micromanipulation systems. Through the structure of a microrobotic system, a microvision system provides a direct way to transmit feedback information. The image information system helps obtain sufficient information for the micromanipulation system, while also compensating for errors resulting from the inaccuracy of the motion model in the micromanipulation system itself. The use of autofocusing, microvision, and automatic control enables automation of the micromanipulation system, eliminating the need for human intervention. This will bring about considerable manpower savings and economic benefits to facilitate mass industrial production. The application of microscopic vision techniques in micromanufacturing, genetic engineering, and nanosensor assembly will make a significant difference to future explorations of human health, drug screening, and cancer, especially in the automatic detection of primary cancer cells.

## Figures and Tables

**Figure 1 micromachines-10-00843-f001:**
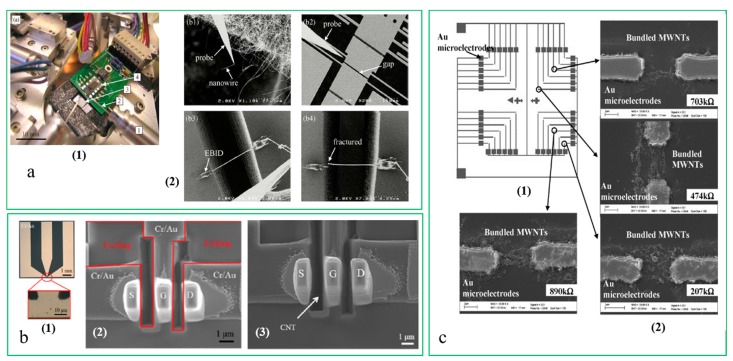
Nanowire transfer. (**a**) (1) Experimental setup; (2) The processing of microassembly (courtesy of [20]); (**b**) Fabrication of three W electrode pillars; (1) Optical microscopy image of Cr/Au outer leads; (2) Achievement of the 3D carbon nanotube (CNT) assembly (courtesy of [24]); (**c**) (1) Drawing of the design for the chip with arrays of Au microelectrodes; (2) SEM images showing the formation of multiwalled carbon nanotubes (MWNTs) between different pairs of Au microelectrodes (courtesy of [13]).

**Figure 2 micromachines-10-00843-f002:**
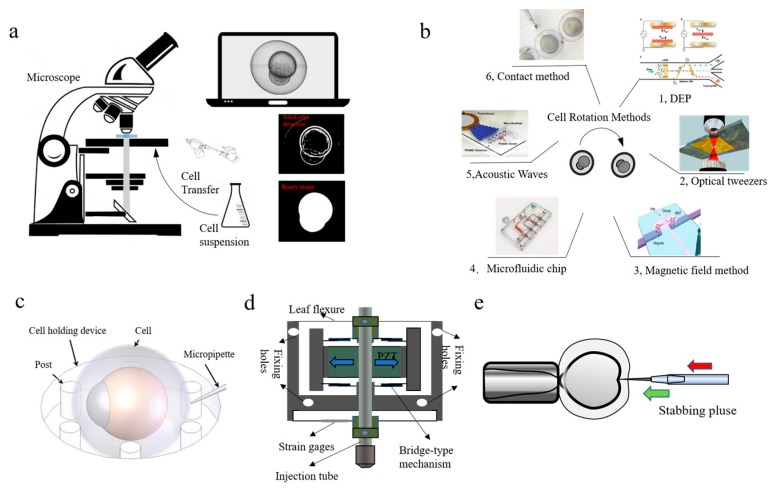
Main parts of a microinjection system. (**a**) Cell detection; (**b**) Cell posture adjustment; (**c**) Schematic configuration of vision-based cellular force measurement (courtesy of [30]); (**d**) Needle actuator (courtesy of [31,32]); (**e**) Cell injection (courtesy of [33]).

**Figure 3 micromachines-10-00843-f003:**
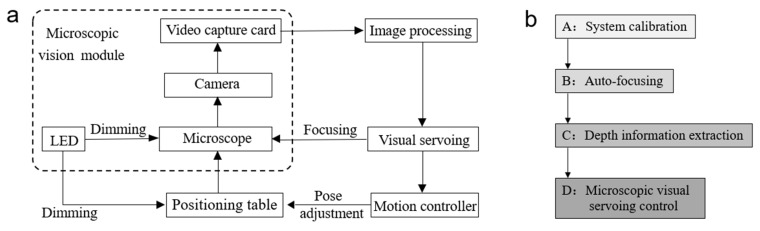
(**a**) The architecture of a microvision system. (**b**) The general process of microscopic visual serving.

**Figure 4 micromachines-10-00843-f004:**
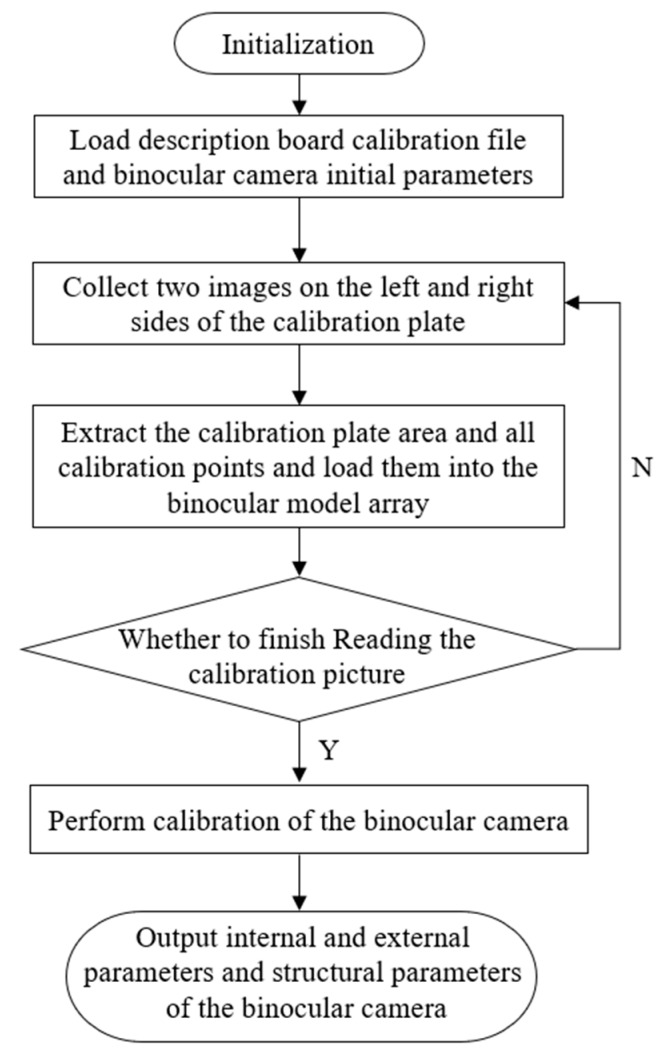
The process of calibrating a microstereoscopic visual system.

**Figure 5 micromachines-10-00843-f005:**
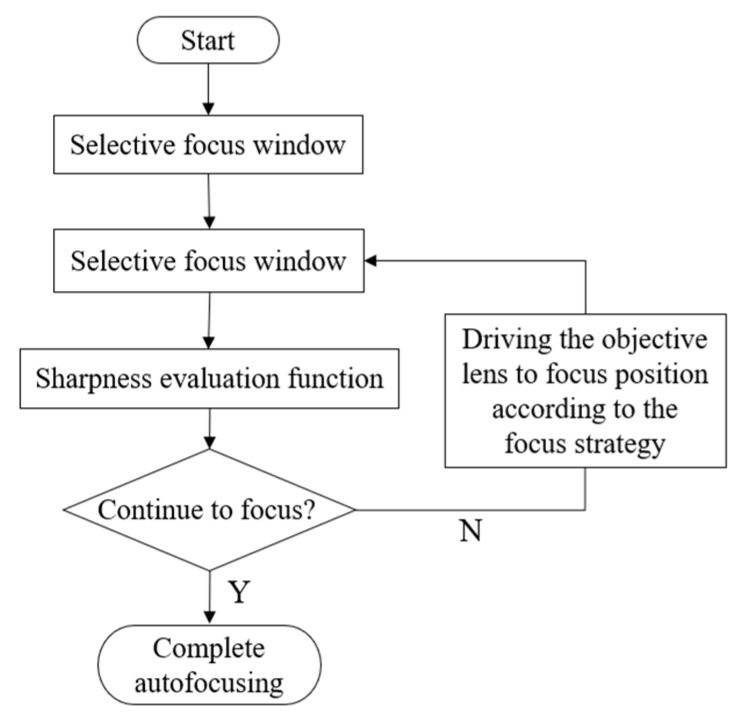
How autofocusing technology works.

**Figure 6 micromachines-10-00843-f006:**
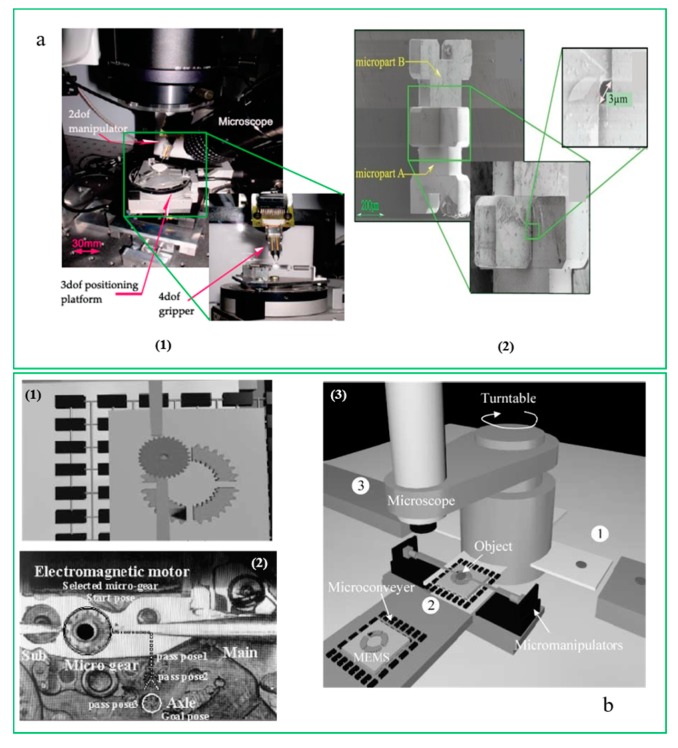
The micromanipulation system in FEMTO-ST. (**a**) (2) shows how a microassembly task is completed (courtesy of [70]); Visual servoing teleoperated microassembly assisted by virtual reality techniques is shown in (**b**) (courtesy of [72]).

**Figure 7 micromachines-10-00843-f007:**
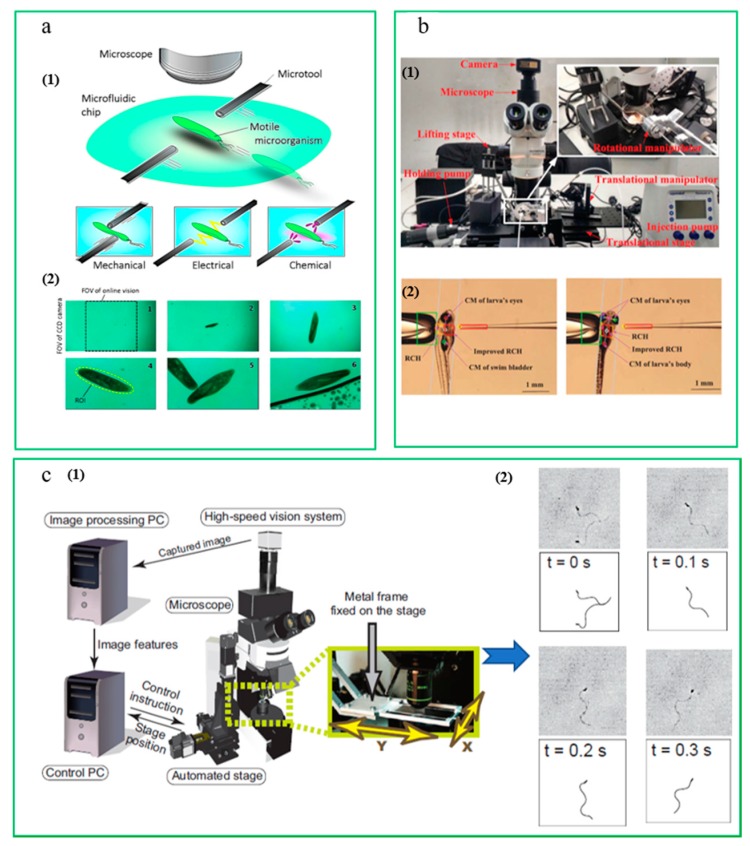
(**a**) Conceptual image of a proposed microrobotic platform. (courtesy of [75]); (**b**) (1) The semiautomated and simple-structure system for Zebrafish larva heart microinjection, (2) Injection phases of the zebrafish embryo. (courtesy of [79]); (**c**) (1) Micromanipulation system in the University of Tokyo; (2) Sequential images and illustrations of a swimming spermatozoon (courtesy of [74]).

**Figure 8 micromachines-10-00843-f008:**
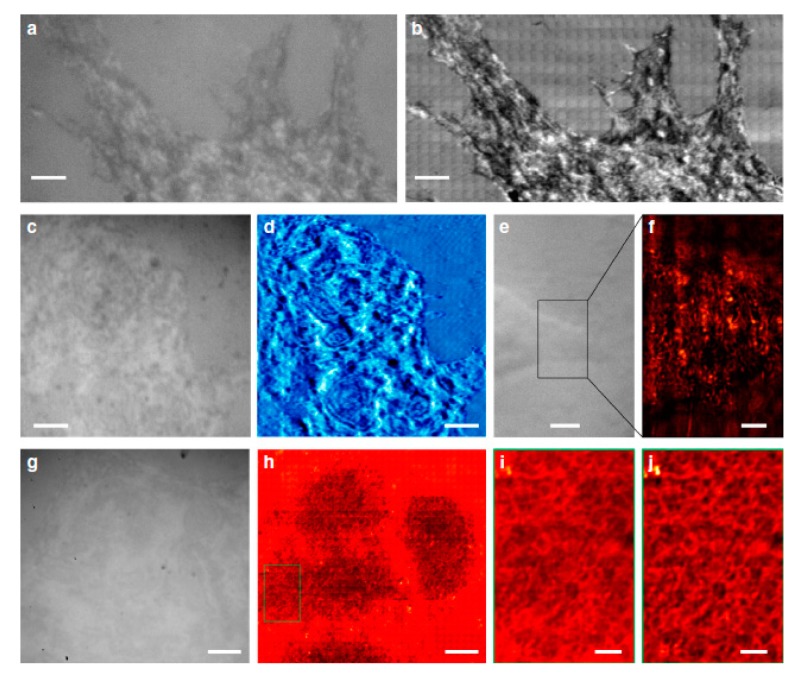
Non-invasive observation of cells in white-light mode. A C2C12 cell was imaged using (**a**) a traditional optical microscope or (**b**) scanning superlens microscopy (SSUM). A video recorded while scanning a C2C12 cell is provided as Supplementary Movie 2. MCF-7 cells were observed (**c**,**e**,**g**) without and (**d**,**f**,**h**) with the aid of the microsphere superlens. A × 100 (numerical aperture (NA) = 0.8) objective was used in (**a**,**b**,**g**,**h**), and a × 50 (NA = 0.6) objective was used in (**c**–**f**). (**i**) Local zoomed area of the marked area shown in (**h**). (**j**) After using a band-pass filter algorithm of (**i**). Scale bars: 6 μm (**a**,**b**); 10 μm (**c**–**e**,**g**,**h**); 3 μm (**f**); 2μm (**i**,**j**) (courtesy of [90]).

**Figure 9 micromachines-10-00843-f009:**
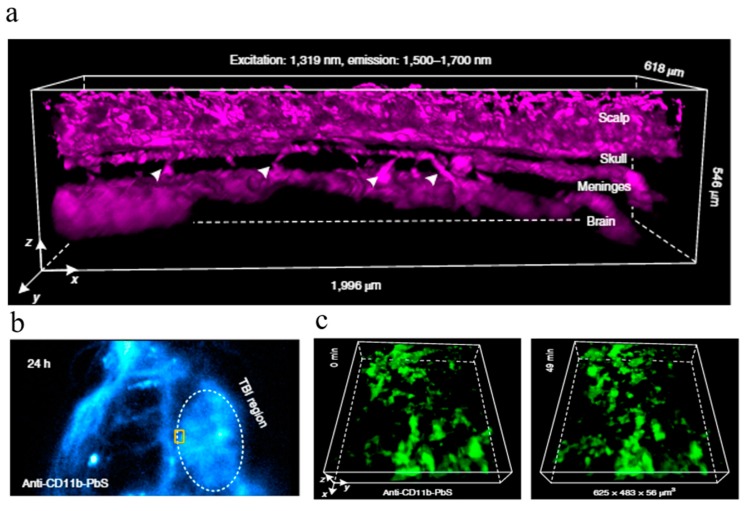
Non-invasive in vivo light-sheet imaging of a mouse head using oblique NIR-II LSM. (**a**) A 3D reconstructed image of blood vessels in an intact mouse visualized through the scalp, skull, meninges and brain cortex, obtained 2 h after intravenous injection of PEGylated PbS/CdS core/shell quantum dot (CSQDs) by oblique NIR-II LSM, as shown in (**b**). (**c**) shows the 3D time-course light-sheet imaging and monitoring of the dynamics of meningeal macrophages and microglia after brain injury, 24 h after the injection of anti-CD11b PEGylated PbS/CdS CSQDs at the boundary of the traumatic brain injury (TBI) region (courtesy of [93]).

**Table 1 micromachines-10-00843-t001:** The methods to obtain the depth information of a microvisual system.

Method	Operational Principles	Advantages	Disadvantages	References
Distance measurement	By installing a laser or ultrasonic generator on the optical microscope, the image information of a certain point can be fused with the information obtained by the laser or ultrasonic generator to get the depth information of object points corresponding to the image points on the controlled objects.	Easy to operate No additional equipment required	Additional equipment required High cost Limited accuracy	[52]
Focus transformation	Record the position of the optical microscope in the optical axis direction at the time when the observed object is clearly imaged, which is the depth information of the system.	Easy to operate No additional equipment required	Limited accuracy Time-consuming	[53,54]
Stereoscopic vision	Placing a microscope in both the horizontal and vertical directions, with the microscope in the horizontal direction being used to observe the depth information of an object.	Long working distance Real-time observation	• Need to build a visual model	[55,56,57,58,59,60]
	Using a stereo light microscope (SLM) as a visual sensor, three-dimensional information can be obtained by the principle of stereo matching.			

**Table 2 micromachines-10-00843-t002:** The classification of visual servo control methods.

Classification	Principles	Characteristics	References
Position-based visual servo (PBVS)	The input is the position of the target in 3D space. The controller output v is a set of velocities of the manipulator joints that directly change the velocity of the end effector. Without manipulator joints, the velocity of the microrobot or manipulated object is indirectly adjusted by changing the current or voltage that controls the field.	Controls the movement of the manipulator directly in the cartesian space Microrobot control and visual processing are done separately Needs to calibrate the inverse kinematics equation Large amount of calculation	[68,69,70,71,72,76,80,81,82,83,84,85,86,87]
Image-based visual servo (IBVS)	The input is the image feature vector of the target. Aims to minimize the error between the current features s(t) extracted from microscopy images and the desired feature s* in the image space. The classical proportional controller $v=-\lambda L_{s}^{+}[s(t)-s^{*}]$, v is the controller output, λ is the gain, $L_{s}^{}$ is the image Jacobian matrix, and $L_{s}^{+}$ is the pseudoinverse of $L_{s}^{}$.	Servo error is directly defined in the image feature space No need to estimate the 3D pose Needs to calculate the pseudoinverse of the Jacobian matrix Good robustness	[73,74,75,77,78,79,88,89]

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
