# Peer review of "A Review on Microscopic Visual Servoing for Micromanipulation Systems: Applications in Micromanufacturing, Biological Injection, and Nanosensor Assembly"

_micromachines, 2019, doi:10.3390/mi10120843_

Round 1

Reviewer 1 Report

In this paper, conventionally developed micromanipulation systems with visual servoing techniques are reviewed and summarized. The general writing and paper organization are well, but, from the reviewer point of view, there are several points which should be addressed to improve the quality of the paper for journal readers.

Comments:

1) Definition of “Micromanipulation” is unclear. Many untethered tools (micro/nano robots) have been widely developed for manipulating small objects and cells. Even through the reviewer understands that this paper is especially focusing on control and applications of large manipulators with tiny tools (e.g., glass), it is better to more clarify the target of this review paper in the introduction part to avoid misleading of readers.

2) In pages 11-14, conventional approaches with micro visual servoing (feedback control) are explained. But, here the authors are just introducing the conventionally developed system in order. These works should be categorized by technical characteristics (contributions), not by research team.

3) Several papers on microscopic visual servoing (feedback) are missed. Please carefully find related works.

For example:

Kim et al., Analysis of rotational flow generated by circular motion of an end effector for 3D micromanipulation, ROBOMECH Journal, 4(5): 1251, 2017. Kim et al., Design of MEMS vision tracking system based on a micro fiducial marker, Sensors and Actuators A: Physical, 234(1): 48-56, 2015. Ahmad et al., Microrobotic platform for single motile microorganism investigation, Micromachines, 8(10): 295, 2017.

4) In section 4, forecast of the applications of micromanipulation with visual servo control are discussed. In addition, the reviewer just let you know that recently visual servoing technique is actively introduced to the bio-imaging field to take single cell level florescent image of moving target.

For example:

Kim et al., Pan-neuronal calcium imaging with cellular resolution in freely swimming zebrafish, Nature Methods, 14: 1107–1114, 2017.

Author Response

Response to the Comments

Thank you for your comments concerning our manuscript entitled “A Review on Microscopic Visual Servoing for Micromanipulation Systems: Applications in Micromanufacturing, Biological Injection, and Nanosensor Assembly”. Firstly we would like to express our sincere thanks to you for your constructive comments and suggestions. Taking into account the comments, we have revised the manuscript. A list of changes that have been made is given in as followings. The comments really help us to study more deeply and improve the quality of the manuscript. For the convenience, we have highlighted the changes to our manuscript by using colored texts in the revised paper. We hope, with these modifications and improvements based on your comments, the quality of our manuscript would meet the publication standard of.

Comment 1: Definition of “Micromanipulation” is unclear. Many untethered tools (micro/nano robots) have been widely developed for manipulating small objects and cells. Even through the reviewer understands that this paper is especially focusing on control and applications of large manipulators with tiny tools (e.g., glass), it is better to more clarify the target of this review paper in the introduction part to avoid misleading of readers.

Response: Thank you very much for your comment. We have redefined the “Micromanipulation” in the introduction part of the article, and clarified the research objects of this review. The specific corrections are highlighted in yellow in this article.

Comment 2: In pages 11-14, conventional approaches with micro visual servoing (feedback control) are explained. But, here the authors are just introducing the conventionally developed system in order. These works should be categorized by technical characteristics (contributions), not by research team.

Response: Thank you very much for your comment. We have categorized by technical characteristics with visual servo control technology. The visual servo control have been divided into position-based visual servo control and image-based visual servo control according to feedback signals, and added corresponding explanations and tables, in which the modified part is marked with yellow highlights in the revised manuscript.

Comment 3: Several papers on microscopic visual servoing (feedback) are missed. Please carefully find related works. For example: Kim et al., Analysis of rotational flow generated by circular motion of an end effector for 3D micromanipulation, ROBOMECH Journal, 4(5): 1251, 2017. Kim et al., Design of MEMS vision tracking system based on a micro fiducial marker, Sensors and Actuators A: Physical, 234(1): 48-56, 2015. Ahmad et al., Microrobotic platform for single motile microorganism investigation, Micromachines, 8(10): 295, 2017.

Response: Thank you very much for your suggestion. We read these literature carefully, and added “Kim et al., Analysis of rotational flow generated by circular motion of an end effector for 3D micromanipulation, ROBOMECH Journal, 4(5): 1251, 2017.” to the introduction part of the article, introduced reference “Design of MEMS vision tracking system based on a micro fiducial marker, Sensors and Actuators A: Physical, 234(1): 48-56, 2015.” to the introductory part of MEMS in the introduction, and added reference “Ahmad et al., Microrobotic platform for single motile microorganism investigation, Micromachines, 8(10): 295, 2017.” to the visual servo control part of the article. And the contents of their citations are elaborated in detail.

Comment 4: In section 4, forecast of the applications of micromanipulation with visual servo control are discussed. In addition, the reviewer just let you know that recently visual servoing technique is actively introduced to the bio-imaging field to take single cell level florescent image of moving target. For example: Kim et al., Pan-neuronal calcium imaging with cellular resolution in freely swimming zebrafish, Nature Methods, 14: 1107–1114, 2017.

Response: Thank you again for your suggestion. We read these literature carefully, and added “Kim et al., Pan-neuronal calcium imaging with cellular resolution in freely swimming zebrafish, Nature Methods, 14: 1107–1114, 2017.” to the conclusion part of the article. We have already marked it with a yellow highlight in the revised manuscript.

 The revised manuscript as the attachment upload.

Reviewer 2 Report

This paper reviewed the developments and applications of micro-vision servoing techniques in micromanipulation including micro stereoscopic vision, visual servoing, geometric calibration, and autofocus techniques. The paper covered most of the critical works done by the leading researchers in the world. Overall, the paper is well-structured and well-written.

Some comments are:

The sequence numbers are not correct after heading 2. The corresponding items in Table 1 are confusing and should be explained more clearly. All the figures should redraw in a high-resolution manner. If the figure came from other authors, do please get the copyrights from them. In Eqs. 1 and 3, the symbol and  are not clearly defined.

Author Response

Response to the Comments

Thank you for your comments concerning our manuscript entitled “A Review on Microscopic Visual Servoing for Micromanipulation Systems: Applications in Micromanufacturing, Biological Injection, and Nanosensor Assembly”. Firstly we would like to express our sincere thanks to you for your constructive comments and suggestions. Taking into account the comments, we have revised the manuscript. A list of changes that have been made is given in as followings. The comments really help us to study more deeply and improve the quality of the manuscript. For the convenience, we have highlighted the changes to our manuscript by using colored texts in the revised paper. We hope, with these modifications and improvements based on your comments, the quality of our manuscript would meet the publication standard of.

Response to Reviewer

This paper reviewed the developments and applications of micro-vision servoing techniques in micromanipulation including micro stereoscopic vision, visual servoing, geometric calibration, and autofocus techniques. The paper covered most of the critical works done by the leading researchers in the world. Overall, the paper is well-structured and well-written. Some comments are:

Comment 1: The sequence numbers are not correct after heading 2. The corresponding items in Table 1 are confusing and should be explained more clearly. All the figures should redraw in a high-resolution manner. If the figure came from other authors, do please get the copyrights from them. In Eqs. 1 and 3, the symbol and are not clearly defined.

Response: Thank you very much for your comment. We are very sorry for our negligence of the incorrect heading sequence numbers in the article. So, we have revised the heading sequence numbers. We have reorganized and revised the contents of the Table 1. The modified Table 1 is as follows and all the figures have been redrawn in high resolution in the article. And obtained the image copyright of the relevant author. We have detailed all the symbols involved in Eq. 1 and Eq. 3 in the article and marked them with yellow highlights. Thank you again for your suggestion.

Table 1. The methods to obtain the depth information of a micro visual system

Method

Operational Principles

Advantages

Disadvantages

References

Distance measuring

By installing a laser or ultrasonic generator on the optical microscope, the image information of a certain point can be fused with the information obtained by the laser or ultrasonic generator to get the depth information of object points corresponding to the image points on the controlled objects

•Easy to operate

•No additional equipment required

•Additional equipment required

•High cost

•Limited accuracy

[52]

Focus

transformation

Record the position of the optical microscope in the optical axis direction at the time when the observed object is clearly imaged, which is the depth information of the system.

•Easy to operate

•No additional equipment required

•Limited accuracy

•Time-consuming

[53,54]

Stereoscopic vision

Placing the microscope in both horizontal and vertical directions, with the microscope in the horizontal direction being used to observe the depth information of an object.

•Long working distance

•real-time observation

•Need to build a visual model

[55–59]

[60]

Using a stereo light microscope (SLM) as a visual sensor, and three-dimensional information can be obtained by the principle of stereo matching.

Round 2

Reviewer 1 Report

The revised paper is well revised based on the comments from the reviewer.